# Fire Resistance, Thermal and Anti-Ageing Properties of Transparent Fire-Retardant Coatings Modified with Different Molecular Weights of Polyethylene Glycol Borate

**DOI:** 10.3390/polym13234206

**Published:** 2021-11-30

**Authors:** Long Yan, Xinyu Tang, Xiaojiang Xie, Zhisheng Xu

**Affiliations:** Institute of Disaster Prevention Science and Safety Technology, School of Civil Engineering, Central South University, Changsha 410075, China; ylong015@csu.edu.cn (L.Y.); TXYtangxinyu0124@163.com (X.T.); xiexiaoj524@163.com (X.X.)

**Keywords:** polyethylene glycol borate, transparent fire-retardant coatings, fire resistance, ageing resistance, thermal stability

## Abstract

Four kinds of polyethylene glycol borate (PEG-BA) with different molecular weights were grafted into cyclic phosphate ester (PEA) to obtain flexible phosphate esters (PPBs), and then applied in amino resin to obtain a series of transparent intumescent fire-retardant coatings. The comprehensive properties of the transparent coatings containing different molecular weights of PEG-BA were investigated by various analytical instruments. The transparency and mechanical analyses indicate that the presence of PEG-BA slightly decreases the optical transparency of the coatings but improves the flexibility and adhesion classification of the coatings. The results from fire protection and cone calorimeter tests show that low molecular weight of PEG-BA exerts a positive flame-retarded effect in the coatings, while high molecular weight of PEG800-BA behaves against flame-retarded effect. Thermogravimetric and char residue analyses show that the incorporation of low molecular weight of PEG-BA clearly increases the thermal stability and residual weight of the coatings and generates a more compact and stable intumescent char on the surface of the coatings, thus resulting in superior synergistic flame-retarded effect. In particular, MPPB1 coating containing PEG200-BA exerts the best flame-retarded effect and highest residual weight of 36.3% at 700 °C, which has 57.6% reduction in flame spread rate and 23.9% reduction in total heat release compared to those of MPPB0 without PEG-BA. Accelerated ageing test shows that low molecular weight of PEG-BA promotes to enhance the durability of structural stability and fire resistance of the coatings, while PEG800-BA with high molecular weight weakens the ageing resistance. In summary, the fire-resistant and anti-ageing efficiencies of PEG-BA in the coatings depend on its molecular weight, which present the order of PEG200-BA > PEG400-BA > PEG600-BA > PEG800-BA.

## 1. Introduction

The use of intumescent fire-retardant coatings is one of the most promising methods to reduce the fire hazard of wood and wood-based products. Normally, the intumescent coatings are composed of a proper binder and the combinations of an acid source, a gas source and a carbon source, which can form a cellular expansion layer that acts as a physical barrier against heat and mass transfer when exposed to flame and high temperature [1]. Currently, transparent intumescent fire-retardant coatings are considered to be a high-efficient material for protection of ancient buildings, cultural relics and high-quality wood-based furniture against fire with reserving their original appearance [2]. However, fabricating transparent fire-retardant coatings faces a huge challenge as both super optical transparency and fire resistance are usually competitive properties [3]. In particular, the traditional flame retardants for the realization of fire resistance usually cause a serious light scattering phenomenon due to the different refractive indices of polymeric matrix and flame retardants, thereby leading to the loss of transparency [4]. The design of transparent fire-retardant coatings based on a reactive-type flame retardant and an active polymer is a promising method to balance both the fire resistance and optical transparency of the coatings [5].

Recently, organophosphate flame retardants have been demonstrated to be a promising candidate of reactive-type flame retardants for fabricating transparent flame-retarded materials [6]. During combustion, organophosphate flame retardants can generate phosphoric acid and its derivatives that catalyze the char formation, thus effectively separating the heat and mass transfer between the combustion zone and underlying materials [7]. In the gas phase, organophosphate flame retardants release P-containing radicals including PO•, HPO• and HPO_2_• that capture H• and HO• radicals from the pyrolysis of polymeric matrix, thus reducing the intensity of combustion reaction [8]. It is well known that some organic compounds and other flame-retarded elements (e.g., silicon, nitrogen, boron, etc.) could improve the flame-retarded efficiency of organophosphate via the synergistic effect [9,10]. Hu et al. [11] designed a high synergistic hyperbranched phosphorus/nitrogen-containing flame retardant (HPNFR), which endows epoxy resin with high fire safety and excellent light transmittance of 85%. Shi et al. [12] synthesized a silicon-containing epoxy/PEPA phosphate flame retardant (EPPSi), and found that the synergism between phosphorus and silicon greatly enhances the fire protection and char formation of transparent fire-retardant coatings. Tian et al. [13] synthesized silane-modified polyphosphate esters via the reaction of dimethyldichlorosilane and polyphosphate ester, then mixed with amino resin to obtain the transparent fire-retardant coatings with the optimum fire-resistant time of 171 min. Shree et al. [2] found that cycloaliphatic epoxy-based phosphate resin acid (PRA) imparts better mechanical and fire-resistant properties to amino transparent fire-retardant coatings compared to both aliphatic epoxy-based PRA and aromatic epoxy-based PRA. Ma et al. [14] synthesized a novel phosphorus-containing flame retardant diallyl 4-phenylphosphonicbenzoic (DAPB) and applied to prepare UV-cured transparent intumescent coating for wood substrates, and found that the coated wood has a self-extinguishing ability. Chen et al. [15] designed DOPO-cellulose acrylate via a two-step reaction of 9, 10-dihydro-9-oxa-10-phosphaphenanthrene-10-oxide (DOPO), cellulose and acryloyl chloride, and applied to fabricate a UV-cured transparent fire-retardant coating with superior anti-dripping property. Moreover, the fire resistance of transparent fire-retardant coatings can be improved significantly by adding a tiny amount of nanoparticles as synergists, and these nanoparticles include layered silicates [16], graphene oxide [17], nano-silica [18]. The uniform dispersion of nanoparticles in the matrix is considered to be a key factor to minimize the light scattering phenomenon of nanoparticles in the matrix and retain the high transparency of the transparent coatings.

Recently, waterborne transparent fire-retardant coatings have attracted large attentions due to their outstanding comprehensive properties and excellent environmental protection effect. However, most of organophosphate with cycle structure is not entirely suit to prepare waterborne transparent fire-retardant coatings due to the poor water solubility [19]. The introduction of water-soluble oligomer into organophosphate is considered as an effective method to improve the water solubility of organophosphate. Shi et al. [20] introduced polyethylene glycol 200 into pentaerythritol phosphate (PEPA) to synthesize high water solubility of PEPA-containing polyether flame retardants for waterborne transparent fire-retardant coatings. The results found that the synergistic effect between PEG and PEPA in waterborne transparent coatings depends on the molecular weight of PEG, which presents the order of PEG200 > PEG150 > PEG400 > PEG600. The presence of PEG is also beneficial to enhance the flexibility of waterborne transparent fire-retardant coatings, as supported by Hong [21]. Apart from PEG chain, polypropylene glycol (PPG) chain could improve the water solubility of PEPA, while poly (tetrahy-drofuran) decreases the water solubility of PEPA [22]. In our previous studies, polyethylene glycol borate (PEG-BA) was introduced into cyclic phosphate ester (PEA) to obtain flexible phosphate esters with high water solubility, and found that a proper content of PEG-BA can significantly improve the fire resistance and smoke suppression properties of waterborne transparent fire-retardant coatings [23]. Although the PEG and PPG chains enhance the fire resistance and water solubility of organophosphates, organic compounds are prone to degrade after exposure of various ageing conditions (e.g., UV radiation, moisture, temperature, chemical solvent, physical stress, etc.), resulting the performance degradation of the coatings [24].

Nowadays, the fire resistance of transparent fire-retardant coatings were evaluated based on newly manufactured samples via various fire standard tests. However, long-term ageing conditions will impair the fire resistance of transparent fire-retardant coatings in the lifetime of the coatings [25]. Because fire protection requirement of transparent fire-retardant coatings may last many years, it is necessary to investigate the ageing behavior of transparent fire-retardant coatings under exposure to ageing conditions. A considerable studies have been devoted to study the ageing resistance and ageing mechanism of opaque intumescent fire-retardant coatings, and the ageing mechanism of the coatings is ascribed to the breakage, degradation and crosslinking of weak bonds in the coating structure and the migration of flame retardants [26]. Moreover, the ageing resistance of a material depends on both the type of ageing factors and its degradation mechanism of the molecular structure under exposure of ageing conditions. The opaque intumescent coatings and transparent intumescent coatings have obvious differences in flame retardant and polymer, resulting in different degradation mechanism and ageing behavior under ageing conditions. To date, relatively few efforts have focused on the ageing behavior of transparent fire-retardant coatings, especially the influence of molecular weight of PEG-BA on the long-term fire resistance of transparent fire-resistant coatings.

In this work, four kinds of PEG-BA with different molecular weight were grafted into cyclic phosphate ester (PEA) to obtain a series of flexible phosphate esters (PPBs), and carefully characterized by FTIR and ^1^H NMR spectra. Then, PPBs were mixed with amino resin to obtain a series of transparent fire-retardant coatings. The influence of PEG-BA molecular weight on the optical transparency, thermal stability, mechanical properties, fire resistance and ageing resistance of the coatings was investigated by various analytical instruments, and the possible flame-retardant and anti-ageing mechanisms of PEG-BA in the coatings is proposed.

## 2. Experimental

### 2.1. Materials

Boric acid (BA, purity: 99.5%), n-butyl alcohol (n-BA) and four kinds of polyethylene glycol with molecular weights of 200, 400, 600 and 800 (PEG200, PEG400, PEG600 and PEG800) were obtained from Sinopharm Chemical Reagent Co., Ltd., Shanghai, China. Phosphoric acid (PA, 85.0% in water) and pentaerythritol (PER, purity: 99.5%) were supplied by Shandong Usolf Chemical Science Co., Ltd., Shandong, China. A typical amino resin namely melamine formaldehyde resin (MF, product number: 303-80, 58–62% in n-BA) was supplied by by Jiyang Sanqiang Chemical Reagent Co., Ltd., Shandong, China. All reagents are used as received without further purification.

### 2.2. Synthesis of Polyethylene Glycol Borate (PEG-BA)

Four kinds of polyethylene glycol borate (PEG-BA) with different molecular weights were synthesized by the esterification reaction of BA and PEG with different molecular weights according to the method as reported [23]. The synthesis route of PEG-BA is presented in Figure 1. Firstly, polyethylene glycol (0.3 mol) and boric acid (6.2 g, 0.1 mol) was added into a 500 mL three-neck round flask. Afterwards, the three-neck flask was removed in an oil bath for magnetic stirring 3 h at 130 °C to obtain the crude product. Then, the crude product was cooled and concentrated under reduced pressure to obtain the PEG-BA. Four kinds of PEG (PEG200, PEG400, PEG600 and PEG800) were used and the obtained PEG-BA products were marked as PEG200-BA, PEG400-BA, PEG600-BA and PEG800-BA, respectively.

### 2.3. Synthesis of Flexible Phosphate Esters (PPBs)

A series of flexible phosphate esters (PPBs) were synthesized by the esterification reaction of PEG-BA and cyclic phosphate ester (PEA) with a mass ratio of 20:80. The synthesis route of PPBs is shown in Figure 2 and described as follows. Firstly, the PEA was synthesized by the reaction of PER, PA and n-BA with a molar ratio of 3:0.85:0.5 according to the procedure as described in our previous study [18]. Afterwards, 80 g PEA and 20 g PEG-BA were added in a 500 mL three-necked flask and the mixture was stirred at 50 °C for 1 h and 115 °C for 4 h. Finally, the target flame retardant namely PPB was obtained via the cooling and concentration under reduced pressure. Four kinds of PEG-BA with different molecular weights (PEG200-BA, PEG400-BA, PEG600-BA and PEG800-BA) were used and the obtained PPBs was designated as PPB1, PPB2, PPB3 and PPB4, respectively. The acid number values of PEA and PPBs were tested according to ASTM D664-11a (2017) standard, and the results are listed in Table 1.

### 2.4. Preparation of Amino Transparent Fire-Retardant Coatings

A series of amino transparent fire-retardant coatings were prepared by mixing 100 g PEA or PPBs ethanol solution (60 wt%) and 120 g MF (58–62 wt% in n-BA), where PEA and PPBs act as curing agent and flame retardant in the coatings. After 45 min’ standing, the obtained coatings were coated on the plywood boards (100 mm × 100 mm × 4 mm, 150 mm × 150 mm × 4 mm and 600 mm × 90 mm × 4 mm) at a wet density of 500 g/m^2^, and the coatings were cured to 0.4 ± 0.02 mm thick film. The coating was applied on the plywood boards with dimension of 300 mm × 150 mm × 4 mm at a wet density of 250 g/m^2^, and the thickness of dry film was 0.2 ± 0.02 mm. The coatings obtained from PEA and PPB1-PPB4 are marked as MPPB0 and MPPB1-MPPB4, respectively.

### 2.5. Characterization

Fourier transform infrared (FTIR) spectra were recorded on an iCAN9 FTIR spectrometer (Tianjin Nengpu Technology Co., China) using KBr pellets in the range of 4000-500 cm^−1^.

^1^H nuclear magnetic resonance (^1^H NMR) spectra of PEA and PPBs was recorded on a Bruker Ascend 400 MHz NMR spectrometer (Bruker, Fällanden, Switzerland) using D_2_O as solvent.

Optical transparency analysis was performed on an LS116-type light transmittance meter (Shenzhen Linshang Technology Co. Ltd., Shenzhen, China). The coatings were coated on the transparent glass slide with a size of 76 mm × 26 mm at a wet density of 250 g/m^2^, and cured to 0.2 ± 0.02 mm thick film.

Optical digital images of the coatings applied on plywood boards were observed on a 52-01000-type optical microscope (Bresser, Bolken, Germany).

Pencil test was applied to measure the pencil hardness of the coatings according to ISO 15184-2012 on a QHQ-A pencil hardness tester (Shenzhen Huaxin Metrology and Detection Technology Co. Ltd., Shenzhen, China). Tape test was conducted to measure the adhesion classification of the coatings according to ASTM D3359-09 on a QFH-A cross-cut adhesive tester (Pushen Testing Instruments Co. Ltd., Shanghai, China). In the pencil and tape tests, the coatings was applied on the plywood boards (100 mm × 100 mm × 4 mm) with a thickness of 0.4 ± 0.02 mm.

Thermogravimetric (TG) analysis was carried on a Q50 instrument (TA instruments, Linden, UT, USA). About 2 mg sample was placed on a alumina pan and then heated from 25 °C to 800 °C at a heating rate of 10 °C/min under nitrogen flow of 40 mL/min.

Cabinet method test was conducted to record the weight loss, char index and intumescent factor of the coatings applied on wood substrates according to ASTM D1360-2011 by an XSF-1-type fire-resistant paint tester (small room mode) (Jiangning Analysis Instrument Company, Nanjing, China) with a sample dimension of 300 mm × 150 mm × 4 mm.

Tunnel method test was conducted to measure the flame spread rate (FSR) of the samples according to ASTM D3806-2011 by a SDF-2-type 2-foot flame tunnel instrument (Jiangning Analysis Instrument Company, Nanjing, China) with a sample dimension of 600 mm × 90 mm × 4 mm.

Heat insulation test was conducted to record the backside temperature of the samples by a cone calorimeter (Fire Testing Technology, East Grinstead, UK) under an external heat flux of 50 kW/m^2^. In the test, the sample coated with transparent coatings was horizontally exposed to the heat radiator with a distance between the coating and radiator bottom is 25 ± 1 mm.

Cone calorimeter test was applied to measure the heat release rate (HRR) and total heat release (THR) of the coatings by a cone calorimeter (Fire Testing Technology, East Grinstead, UK) according to ISO5660-2002 standard. Each specimen with dimension of 100 mm × 100 mm × 4 mm was horizontally exposed to the heat radiator under an external heat flux of 50 kW/m^2^.

The scanning electron microscope (SEM) images of char residues were observed on a MIRA 3 LMU scanning electron microscopy (Tescan, Brno, Czech Republic) under the accelerating voltage of 20 kV. Element analysis of char residues was measured by energy dispersive X-ray spectroscopy (EDS) maps on an X-Max20 X-ray probe (Oxford Instruments, Aylesbury, UK).

Accelerating aging test was performed on an UV weathering chamber (Dongguan HaoRan Testing Instrument Co. Ltd, Guangdong, China) according to ASTM G154-2006 standard. The samples were exposed to the cycles of UV irradiation and condensation for regular time intervals. Each cycle had an 8 h UV irradiation with the black panel thermometer temperature of 60 ± 3 °C and 4 h condensation with the black panel thermometer temperature of 50 ± 3 °C. The wavelength of radiation was 340 nm, and the intensity of UV light was 0.76 W/(m^2^·nm). The samples were taken out from the ageing oven after 2, 6 and 11 cycles for various measurements and characterization.

## 3. Results and Discussion

### 3.1. Chemical Structure Characterization

The FTIR spectra of PEG-BA with different molecular weights are presented in Figure 1. Four kinds of PEG-BA show similar characteristic peaks, and the broad absorption peaks at 3400 and 1647 cm^−1^ are assigned to the–OH groups. The characteristic peaks at 2883, 1420, 1351, 1101 and 677 cm^−1^ are assigned to the stretching vibration of –CH_2_–groups, B–O groups, B–O–C groups, C–O–C groups and BO_3_ groups, respectively [27]. Notably, the new absorption peak of B–O–C groups is appeared at 1351 cm^−1^, verifying that the PEG-BA products were synthesized successfully.

The FTIR spectra of PEA and PPBs were presented in Figure 2. It can be seen that PEA and PPBs show similar characteristic peaks, and the peaks at 3410, 2960, 2887, 2331, 1646, 1466, 990 and 883 cm^−1^ are assigned to –OH stretch vibration, –CH_3_ stretch vibration, asymmetric and symmetric–CH_2_–stretch, P–OH stretch vibration, –OH bend vibration,–CH_2_ deformation vibration, exocyclic P–O–C stretch vibration and cyclic P–O–C stretch vibration, respectively [17]. In the spectra of PPBs, a new absorption peaks at 1350 cm^−1^ belonging to B–O–C stretch vibration is found, indicating PEG-BA was successfully grafted to the structure of PEA. Moreover, the decreased acid number of PPBs further verify the esterification reaction between P–OH groups in PEA and C–OH groups in PEG-BA.

The ^1^H NMR spectra of PEA and PPBs are presented in Figure 3. In the spectrum of PEA, two major signals located at 5.16 ppm and 2.20 ppm are assigned to methylene protons in the cyclic P–O–CH_2_–structure (labelled 4) and methylene protons in the exocyclic P–O–CH_2_–structure (labelled 2, 3), respectively. Moreover, the peak at 0.09 ppm is attributed to the methylene protons in the n-BA. In the spectra of PPBs, the peaks of methylene protons in the exocyclic P–O–CH_2_–structure (labelled 3) exhibit high frequency than that of PEA, which is ascribed to the protons in P–OH groups replaced by the high electronegativity of PEG-BA chain. Additionally, a new peak at 0.25–0.5 ppm is assigned to methylene protons in the PEG chain (labelled 2). All above results confirm that PPBs have been synthesized successfully as shown in Figure 2.

### 3.2. Transparency and Morphology Analyses

The morphology of flame retardants and their resulting coatings are presented in Figure 4. As shown in Figure 4, PEA and the PPBs form a uniform liquid without phase separation, and the transparency of PPBs gradually decreases with the increase of molecular weight of PEG-BA. Especially, PPB4 becomes a yellowish liquid, which is ascribed to the high radius and molecular weight of PEG800-BA result in the significant enhancement of light scattering. From the digital photos of MPPBs coatings, it is found that all the coatings exhibit high light transmittance, and the wood texture is clearly visible after application of coatings. When PEG-BA is introduced, the transparency of the resulting coating is gradually decreased with increasing molecular weight of PEG-BA. In particular, MPPB4 exhibits the lowest transparency of 87.5%, which is still characterized by a high light transmittance. All above results show that the introduction of PEG-BA slightly decreases the light transmittance of the coatings, and low molecular weight of PEG-BA is beneficial to maintain the high transparency of the coatings.

### 3.3. Hardness and Adhesion Analyses

The pencil hardness and adhesion classification of the coatings are presented in Table 2. The pencil hardness of MPPB1-MPPB4 is increased to 3B rating from B rating of MPPB0 coating, indicating the increase of flexibility. All the coatings shows the high adhesion classification above 3B rating, which means excellent adhesion property. Additionally, the adhesion classification of MPPB1 is improved to 4B rating from 3B rating of MPPB0, indicating that the addition of PEG200-BA increases the adhesion property of the coatings. All above results show that the introduction of PEG-BA chain is beneficial to increase the flexibility and maintain the high adhesion of the coatings.

### 3.4. Thermal Stability Analysis

The TG and DTG curves of the MPPBs coatings under nitrogen atmosphere are presented in Figure 5. The related thermal parameters, including the onset decomposition temperature at 5% mass loss (*T*_on_), the temperature at maximum weight loss rate (*T*_max_), peak mass loss rate (*PMLR*) and the residue at 700 °C, are listed in Table 3. In the nitrogen atmosphere, the coatings exhibit four decomposition stages, respectively, during 60–200 °C, 200–300 °C, 300–550 °C, and 550–700 °C, where the third stage is the dominated one. The first stage at 60–200 °C is attributed to the release of small molecules in amino resin and phosphate esters. The second stage at 200–300 °C is assigned to the breakage of C–O–C groups in pentaerythritol phosphate and PEG chain, and MPPB1-MPPB4 coatings show higher mass loss and *PMLR* value than those of MPPB0 due to the pyrolysis of PEG-BA chain in PPBs. The third stage at 300–550 °C is attributed to the decomposition of amino resin and phosphate esters, the phosphate derivatives and ethylene glycol ester chain decomposed from phosphate esters interact with triazine derivatives and inert gases decomposed from amino resin to generate multicellular char. The fourth stage at 550–700 °C is assigned to the decomposition of unstable structures including C–C bonds or ether linkages in the multicellular char. The residue yield at 700 °C firstly increases and then decrease with the addition of PEG-BA, indicating the introduction of low molecular weight of PEG-BA enhances the char forming ability of the coatings. In particular, PEG200-BA imparts the resulting MPPB1 coatings with the highest residual weight of 36.3% at 700 °C among the samples.

It can be seen from Table 3 that the onset decomposition temperature (*T*_on_) of the coatings gradually increases with the increasing molecular weight of PEG-BA, which means the enhanced thermal stability of the coatings. Additionally, the char formation of the coatings firstly increases and then decreases with the increasing molecular weight of PEG-BA, and PEG800-BA behaves against the char formation of the coatings. In summary, the introduction of low molecular weight of PEG-BA contributes to enhance both the thermal stability and char formation of the coatings.

### 3.5. Fire Resistance Analysis

The backside temperature curves of the coatings obtained from the heat insulation test are presented in Figure 6. As seen from Figure 6, the backside temperature of uncoated plywood board rapidly increases and reaches above 220 °C at 180 s. The application of the coatings greatly decreases the backside temperature of plywood board, indicating the enhancement of heat insulation property. From MPPB0 to MPPB4, the equilibrium backside temperature at 900 s is 205.3 °C, 135.6 °C, 157.9 °C, 183.9 °C and 219.1 °C, respectively. It can be seen that the equilibrium backside temperature at 900 s of the coatings firstly decreases and then increases with the increasing molecular weight of PEG-BA, and MPPB4 containing PEG800-BA shows worse heat insulation property than that of MPPB0. This results indicates that the addition of low molecular weight of PEG-BA has a positive effect on enhancing the heat insulation property of the coatings, and the high molecular weight of PEG800-BA behaves against flame-retardant effect. In particular, PEG200-BA imparts the best heat insulation property to the coatings among the samples.

The fire-resistant performance of the coatings obtained from the cabinet method and tunnel method tests are listed in Table 4. As seen from Table 4, the application of the coatings greatly enhances the fire safety of the wood substrates. When PEG-BA is introduced, the fire-resistant performance of the coatings firstly increases and then decreases as the increasing molecular weight of PEG-BA, and PEG800-BA decreases the fire-resistant performance of the coating. It reveals that the low molecular weight of PEG-BA exerts a positive effect on enhancing the fire-resistant performance of the coatings, and PEG800-BA has a negative effect on the fire-resistant performance. These results are consistent with the results of heat insulation property.

The digital photographs and SEM images of the char residues after the cabinet method test are presented in Figure 7. Obviously, the introduction of low molecular weight of PEG-BA increases the compactness and intumescence of the char residues, and MPPB1 shows the highest char layer with the intumescent factor of 75.5. The addition of PEG800-BA decreases the compactness and intumescence of the char, thus resulting in the decrease of fire resistance. As seen from the SEM images, the coatings form a more compact char layer with less voids and cracks as the decreasing molecular weight of PEG-BA, which means the enhanced barrier effect and fire resistance. Generally, the enhanced barrier effect of the char not only prevents the diffusion of combustible gases and heat but also segregates the contact of combustible gases with oxygen, thus strengthens the fire resistance of the coatings. Among the samples, MPPB1 char has least voids and cracks concomitant with the best fire resistance, while MPPB4 shows the biggest voids and cracks concomitant with the worst fire resistance.

The elemental compositions of the char obtained from the cabinet method test are listed in Table 5. It is clearly seen that the content of B and P elements in the char is gradually increased with the decreasing molecular weight of PEG-BA in the coatings, which is ascribed to the content of B element in the coatings is gradually increased with the decreasing molecular weight of PEG-BA. The increased B and P elements is beneficial to form more B–O–C and P–O–C crosslinking structures in the condensed phase. Additionally, the weight ratio of C/O in the char firstly increases and then decreases with the increasing molecular weight of PEG-BA, and MPPB1 char exhibits the highest C/O mass ratio of 2.88 while MPPB4 char shows the lowest C/O mass ratio of 1.67 among the samples. The increased C/O mass ratio is beneficial to improve the cross-linking density as well as barrier effect of the char. In particular, MPPB1 exhibits best barrier effect and fire resistance among the samples, corresponding to the highest C/O ratio and total content of B and P elements in the char. The above results reveal that the addition of low molecular weight of PEG-BA is beneficial to enhance the fire resistance of the coatings, while the high molecular weight of PEG800-BA performs against fire resistance to the coatings.

An illustration of the flame-retardant mechanism of PPBs in the transparent coatings is presented in Figure 3. The flame-retarded effect of PPBs in the coatings acts in both gases phase and condensed phase, and the main flame-retarded effect is ascribed to a strong barrier effect in the condensed phase. The addition of low molecular weight of PEG-BA is beneficial to enhance the cross-linking density, anti-oxidation ability and compactness of the char residues, thus endowing the resulting coatings with super barrier effect and insulation property. However, the presence of PEG800-BA has a negative effect on the barrier formation, thus decrease the barrier effect as well as the fire resistance of the coating.

### 3.6. Cone Calorimetric Analysis

The HRR and THR curves of the transparent fire-retardant coatings under an external heat flux of 50 kW/m^2^ are presented in Figure 8. As seen from Figure 8, the HRR values of the coatings rapidly increase after ignition and appear a strong peak at about 65 s. MPPB0 is ignited at 30 s and reaches a peak heat release rate (PHRR) value at 58 s. When PEG-BA is introduced, the PHRR and THR values of the coatings firstly decrease and then increase with the increasing molecular weight of PEG-BA. In detail, the PHRR value from MPPB0 to MPPB4 is 79.0, 56.5, 64.4, 73.2 and 102.1 kW/m^2^, respectively, and the THR values from MPPB0 to MPPB4 is 33.1, 25.2, 27.2, 30.8 and 46.1 kW/m^2^, respectively. It is clearly shown that the low molecular weight of PEG-BA shows an obvious synergistic flame-retarded effect on reducing the heat release of the coatings, while the high molecular weight of PEG800-BA shows a negative flame-retarded effect in the coatings. In particular, PEG200-BA exerts the best synergistic flame-retarded effect in the coatings, which is consistent with the results from the fire protection tests.

Digital photographs of the MPPBs char collected from the cone calorimeter test are presented in Figure 9. It is clearly seen that the thickness and compactness of the char layer are gradually decreased with increasing molecular weight of PEG-BA, and MPPB1 coating containing PEG200-BA forms the highest and densest char concomitant with the best synergistic effect among the samples. When PEG800-BA is added, the resulting MPPB4 coating forms the lowest and loosest intumescent char, corresponding to the highest HRR and THR values among the samples. The above results indicate that the low molecular weight of PEG-BA can serves as an effective synergist for enhancing the flame retardancy of the coatings, and the high molecular weight of PEG800-BA is not suitable for the transparent coatings.

### 3.7. Ageing Resistance Analysis

The optical micrographs of the coatings under different ageing cycles are shown in Figure 10. The MPPBs coatings show surface damage after 2 ageing cycles, and the surface density of cracks and bubbles increases with the increasing of ageing cycles. The application of low molecular PEG-BA is beneficial to decrease the deterioration degree of coating surface under ageing conditions, and the structural integrity of coatings increases with the decreasing molecular weight of PEG-BA. Especially, MPPB1 containing PEG200-BA is least affected by the accelerated ageing conditions among the samples. Besides, the introduction of PEG800-BA decreases the structural integrity of the coating, and MPPB4 shows weakest structural integrity after 11 ageing cycles among the samples.

Fire-resistant time is defined as the time when the backside temperature of the samples reaches 220 °C in the heat insulation test. The fire-resistant time of the coatings under different accelerated ageing treatment are presented in Figure 11. As seen in Figure 11, the fire-resistant time gradually decreases with increasing ageing cycles due to the weakened structural integrity of the coatings, this in correspondence with the morphology analyses. Compared to MPPB0, MPPB1-MPPB3 is less affected by accelerated ageing process, while MPPB4 is more affected by accelerated ageing process. Especially, MPPB1 shows the lowest deterioration degree of fire-resistant time, while MPPB4 exhibits the highest deterioration degree of fire-resistant time. The above results verify that low molecular weight of PEG-BA is beneficial to endow the coatings with superior durability of fire resistance.

The FTIR spectra of the transparent fire-retardant coatings recorded at different accelerated ageing cycles are presented in Figure 12. The FTIR assignments of main characteristic groups for MPPBs coatings under different ageing treatment are shown in Table 6. It can be seen from Figure 12 that all the coatings show similar characteristic peaks before and after ageing treatment, the intensities of some absorption peaks vary with the ageing cycles. The intensities of –NH_2_ and –OH groups at 3374 cm^−1^ are gradually strengthened with increasing ageing cycles, which is ascribed to the breakage of low-bond-energy C–O and C–N in amine resin generates small molecular amines and alcohols after exposure to UV radiation [28]. The peaks of–CH_2_ groups at 2960 cm^−1^, P–OH groups at 2370 cm^−1^, N–H/C=O groups at 1650 cm^−1^ and C=N groups at 1560 and 792 cm^−1^ almost remain unchanged before and after ageing treatment. The peaks of P=O groups at 1315 cm^−1^ are gradually strengthened with increasing ageing cycles due to the degradation, oxidation and hydrolyzation of P–O–C groups in PPBs flame retardants. The intensities of the overlapping vibration of P–O–C, P–O–P and C–O bonds at 1070 cm^−1^ are gradually increased after ageing treatment, which is ascribed to the degradation and oxidation of pentaerythritol phosphate and PEG chain that generates pyrophosphate and alcohols rich in P–O–P and C–O groups [29,30]. The above results show that the reactions of degradation and oxidization and hydrolyzation change the molecular structure of the coatings after exposure to ageing conditions, resulting in the degradation of structural integrity and fire resistance. Moreover, it is noted that the intensities of the peaks of P=O groups, P–O/C–O groups in the spectra of MPPBs coatings decrease with addition of low molecular weight of PEG-BA after the same ageing treatment, indicating that the introduction of low molecular weight of PEG-BA is beneficial to enhance the structural stability of the coatings. When PEG800-BA is added, the intensities of the peak at 3374 cm^−1^ in the spectrum of MPPB4 is stronger than those of MPPB0 after the same ageing treatment, indicating that the introduction of PEG800-BA decreases the structural stability of amino matrix. This phenomenon is consistent with the results from the morphology analyses and fire-resistant tests of the coatings. Based on the above results, it can be concluded that the introduction of low molecular weight of PEG-BA is beneficial to enhance the structural stability and durability of fire resistance of the coatings, while PEG800-BA plays a negative effect on the ageing resistance of the coatings.

## 4. Conclusions

Four kinds of PEG-BA with different molecular weights were successfully grafted to the structure of PEA to obtain four kinds of flexible phosphate esters (PPBs), and the chemical structures of PPBs were carefully characterized by FTIR and ^1^H NMR spectra. The obtained PPBs were mixed with amino resin to obtain amino transparent fire-retardant coatings, and the influence of molecular weight of PEG-BA on the mechanical properties, fire resistance and anti-ageing behavior of the coatings was assessed by various analytical methods. Transparency analysis shows that the introduction of PEG-BA slightly decreases the transparency of the coatings, and MPPB4 containing PEG800-BA exhibits the lowest transparency of 87.5% which still keeps indeed high transparency. Hardness and adhesion analyses indicate that the existence of PEG-BA is beneficial to increase the flexibility and adhesion classification of the coatings. TG analysis shows that the presence of PEG-BA with low molecular weight is beneficial to enhance the residual weight of the coatings, while PEG800-BA with high molecular weight behaves negative effect on the char formation of the coating. In particular, MPPB1 containing PEG200-BA shows the highest residual weight of 36.3% at 700 °C, while MPPB4 containing PEG800-BA exhibits the lowest residual weight of 26.9% at 700 °C. Fire protection and cone calorimeter tests show that the incorporation of PEG-BA with low molecular weight enhances the flame-retarded effect of the coatings, while high molecular weight of PEG800-BA behaves against flame-retarded effect. Especially, MPPB1 exerts the best flame-retarded effect among the coatings, which exhibits 28.5% reduction in peak heat release rate and 57.6% reduction in flame spread rate compared to those of MPPB0 obtained from PEA. Char residue analysis shows that low molecular weight of PEG-BA contributes to form a more dense, compact and intumescent char against heat and mass transfer, thus effectively imparting the coatings with superior fire resistance. However, high molecular weight of PEG800-BA inhibits the intumescence and char formation of the coating, thus behaving against flame-retarded effect. Accelerated ageing test shows that low molecular weight of PEG-BA has a positive effect on the durability of structural stability and fire resistance of the coatings, while high molecular weight of PEG800-BA behaves against ageing resistance to the coating. The ageing resistance of the coatings decreases with the increasing molecular weight of PEG-BA, and PEG200-BA imparts the superior ageing resistance to the coating via weakening the blistering and powdering phenomenon. In summary, low molecular weight of PEG-BA is more conducive to impart the coatings with superior optical transparency, char formation, fire resistance and ageing resistance.

## Data Availability

The raw/processed data required to reproduce these findings cannot be shared at this time as the data also forms part of an ongoing study.

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
