# Peer review of "Fire Resistance, Thermal and Anti-Ageing Properties of Transparent Fire-Retardant Coatings Modified with Different Molecular Weights of Polyethylene Glycol Borate"

_polymers, 2021, doi:10.3390/polym13234206_

Round 1
Reviewer 1 Report
Presented work is interesting, but should be completed in some points:
- The mechanism of action of prepared boron- and phosphorus-based flame retardants should be described in details (and supported by chemical reactions)
- Obtained results should be discussed with the literature (with works related to the boron- and phosphorus-based flame retardant)
- How did you confirm the thermal decomposition mechanism (Lines 297-313)?
- In my opinion, thermal stability should be also investigated in air atmosphere
- Did you try to measure the thickness of the coatings?
- Figure 1 and Figure 13 (FTIR spectra): The peak at around 2370 cm-1 looks very similar and correspond to the presence of CO2 in atmosphere - Please check: https://doi.org/10.1016/j.aca.2017.03.053
Reviewer 2 Report
This paper describes the preparation of fire-retardant coatings from polyethylene glycol borate, and their fire resistance, thermal and anti-ageing properties. The information presented in this paper seems to be useful for researchers in this area. Minor revisions are required before publication.
At the bottom of page 11, the authors should mention the difference between "voids" and "holes".
At line 2 on page 12, the authors stated that the content of B and P elements in the char is gradually increased with the decreasing molecular weight of PEG-BA in the coatings. The reason for this tendency should be discussed.
At line 4 on page 13, "Figure 10" is a typo of "Figure 9".
The pictures presented in Figure 10 are similar to each other and not informative. I consider these pictures can be omitted.
Round 2
Reviewer 1 Report
Dear Authors,
Thank you very much for your detailed responses and performed completions and corrections. In my opinion presented work can be accepted in present form.